

# A final word on FCNC-Baryogenesis from two Higgs doublets

Wei-Shu Hou[1], Tanmoy Modak[2*] and Tilman Plehn[2]

**1** Department of Physics, National Taiwan University, Taipei, Taiwan
**2** Institut für Theoretische Physik, Universität Heidelberg, Germany

★ modak@thphys.uni-heidelberg.de

## Abstract

Electroweak baryogenesis in a two-Higgs doublet model is a well-motivated and testable scenario for physics beyond the Standard Model. An attractive way of providing $CP$ violation is through flavor-changing Higgs couplings, where the top-charm coupling is hardly constrained. This minimal scenario can be tested by searching for heavy charged and neutral Higgs bosons at the LHC. While the charged Higgs signature requires a dedicated analysis, the neutral Higgs signature will be covered by a general search for same-sign top pairs. Together, they provide a conclusive test of this kind of baryogenesis.

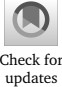

## 1 Introduction

The Higgs discovery [1,2] and subsequent measurements of the Higgs Lagrangian at Run 2 [3–11] indicate that the Standard Model is the correct effective theory around the electroweak

scale. While there exists no experimental evidence for physics beyond the Standard Model so far, extended Higgs sectors are motivated by theoretical considerations, like mass generation of up-type and down-type fermions, neutrino mass generation, electroweak baryogenesis, or dark matter. In particular, two-Higgs doublet models (2HDMs) [12–14] are an integral part of well-defined models for physics beyond the Standard Model, including MSSM [15], composite Higgs models [16], little Higgs models [17, 18], or GUTs [19–22].

If we use baryogenesis [23] as a guiding principle to new physics searches at the LHC, a 2HDM is an attractive and minimal choice. It can provide both, new scalar degrees of freedom [24, 25] and $CP$-violation. In the general [26, 27] or type-III [28] 2HDM, the new particles can be close in mass to the SM-Higgs [29, 30]. Sufficiently large $CP$-violation is non-trivial to achieve, in our model we rely on the Yukawa sector. If both doublets couple to up-type and down-type quarks, they define two separate Yukawa matrices. After diagonalizing the quark mass matrices we find the real, diagonal couplings $\lambda_{ii} = \sqrt{2}m_i/v$ and the complex, non-diagonal couplings $\rho_{ij}$. While flavor-changing neutral couplings are generally well-constrained, it is possible to have electroweak baryogenesis (EWBG) driven by a single, order-one, complex coupling $\rho_{tc}$ [31],

$$\text{Im}\,\rho_{tc} \gtrsim 0.5 \qquad \text{and} \qquad |\cos\gamma| \gtrsim 0.1 \,, \tag{1}$$

where $\gamma$ is the mixing angle between the two $CP$-even Higgs states.

It has been shown [32–36] that the coupling $\rho_{tc}$ can be discovered in the LHC process

$$cg \to tA/H \to t\,(t\bar{c})\,, \tag{2}$$

where this process retains a very mild dependence on $\cos\gamma$ and is especially useful for small values of $\cos\gamma$. To attribute this signal to EWBG requires information on the mixing angle $\cos\gamma$, for instance through $b$-associated charged Higgs production [37]

$$cg \to bH^+ \to b\,(W^+h)\,. \tag{3}$$

Here the production process is induced by $\rho_{tc}$ [38, 39], while the decay amplitude is proportional to the mixing angle $\cos\gamma$. Even in the absence of complex phase information, such a search can test the required particle content and parameter space for the $\rho_{tc}$-EWBG scenario. Finally, the exotic top decay

$$t \to ch \tag{4}$$

is induced by the coupling $\rho_{tc}$ combined with non-vanishing $\cos\gamma$ [28] and is searched for by CMS [40] and ATLAS [41].

In this paper we show how the two LHC searches for charged and neutral heavy Higgs bosons can conclusively probe the parameter region required for $\rho_{tc}$-EWBG in the general 2HDM (g2HDM). The paper is organized as follows: in Sec. 2 we discuss the model and its preferred parameter space, and then compare it to the reach of the charged Higgs channel in Sec. 3. Section 4 is dedicated to same-sign top production from neutral Higgs production and its complementarity to the charged Higgs signature. We summarize our results in Sec. 5.

## 2 Model and parameter space

The general $CP$-conserving two Higgs doublet potential can be written as [42, 43]

$$V(\Phi, \Phi') = \mu_{11}^2 |\Phi|^2 + \mu_{22}^2 |\Phi'|^2 - \left(\mu_{12}^2 \Phi^\dagger \Phi' + \text{h.c.}\right) + \frac{\eta_1}{2}|\Phi|^4 + \frac{\eta_2}{2}|\Phi'|^4 + \eta_3 |\Phi|^2 |\Phi'|^2$$
$$+ \eta_4 |\Phi^\dagger \Phi'|^2 + \left[\frac{\eta_5}{2}(\Phi^\dagger \Phi')^2 + \left(\eta_6 |\Phi|^2 + \eta_7 |\Phi'|^2\right)\Phi^\dagger \Phi' + \text{h.c.}\right]. \tag{5}$$

In the Higgs basis, the VEV $v = 246$ GeV is generated by the doublet $\Phi$, while $\Phi'$ does not develop a VEV, hence $\mu_{22}^2 > 0$. The minimization conditions in the two field directions lead to $\mu_{11}^2 = -\eta_1 v^2/2$ and $\mu_{12}^2 = \eta_6 v^2/2$. The mixing angle $\gamma$ diagonalizes the $CP$-even mass matrix to define the mass eigenstates $h$ and $H$ [42, 43],

$$c_\gamma^2 = \cos^2\gamma = \frac{\eta_1 v^2 - m_h^2}{m_H^2 - m_h^2} \qquad \text{and} \qquad s_{2\gamma} = \sin(2\gamma) = \frac{2\eta_6 v^2}{m_H^2 - m_h^2}, \tag{6}$$

where $c_\gamma \to 0$ in the alignment limit ($\eta_6 = 0$ and $\eta_1 = m_h^2/v^2 \sim 1/4$). To satisfy the first Sakharov condition [23], a new scalar degree of freedom close in mass with the SM-Higgs [44–46] can trigger a strong first-order phase transition. Following Eq.(6) this is guaranteed by finite $c_\gamma$ and perturbatively stable $\eta_i = \mathcal{O}(1)$, for instance $\eta_6 = \mathcal{O}(1)$ and $\eta_1 = \mathcal{O}(1) > m_h^2/v^2$ [43].

Next, baryogenesis requires a complex phase in the Higgs or Yukawa sectors [23]. Many analyses have studied a complex Higgs potential, which tends to be strongly constrained by EDM measurements [47–50]. We look at the alternative option of $CP$-violation arising from the Yukawa sector [31, 42, 51]

$$\mathcal{L} \supset -\frac{1}{\sqrt{2}} \sum_{F=U,D,L} \overline{F}_i \Big[ \left(-\lambda_{ij}^F s_\gamma + \rho_{ij}^F c_\gamma\right) h + \left(\lambda_{ij}^F c_\gamma + \rho_{ij}^F s_\gamma\right) H - i \, \text{sgn}\,(Q_F)\rho_{ij}^F A \Big] P_R F_j$$
$$- \overline{U}_i \big[(V\rho^D)_{ij} P_R - (\rho^{U\dagger}V)_{ij} P_L \big] D_j H^+ - \bar{\nu}_i \rho_{ij}^L P_R L_j H^+ + \text{h.c.}, \tag{7}$$

where $i, j = 1, 2, 3$ are generation indices, $P_{L,R} \equiv (1 \mp \gamma_5)/2$, and $V$ is the CKM matrix. In flavor space, the fermion fields $F$ are defined as $U = (u, c, t)$, $D = (d, s, b)$, $L = (e, \mu, \tau)$ and $\nu = (\nu_e, \nu_\mu, \nu_\tau)$. While the mass matrices are diagonalized as in the Standard Model, one cannot rotate away $CP$-violating phases of the second set of $\rho^F$ matrices in the general 2HDM even in the Higgs basis. That is, the two coupling matrices are

$$\lambda_{ij}^F = \sqrt{2}\,\frac{m_i^F}{v}\delta_{ij} \in \mathbb{R} \qquad \text{and} \qquad \rho_{ij}^F \in \mathbb{C}\,. \tag{8}$$

The complex coupling matrices $\rho^F$ are, strictly speaking, not related to the fermion masses. On the other hand, given experimental constraints and a possible order-of-magnitude correspondence in the values of $\rho^F$ and $\lambda^F$ lead us to consider $\rho_{tj}^U$ or $\rho_{tt}^U$.

In principle, a complex $\rho_{tt}$ can robustly drive EWBG [31], which motivates search for channels like $gg \to H \to t\bar{t}$ or $gg \to Ht\bar{t} \to 4t$ [52–54]. In this paper we focus instead on complex off-diagonal entries $\rho_{tj}$, specifically $\rho_{tc}$. With a large phase, this FCNC coupling can also account for the observed baryon asymmetry [31]. One of its merits is that $\rho_{tc}$ does not generate an electron EDM through the Barr-Zee [55] two-loop mechanism, and can therefore more easily [56] evade the ACME bound [57] $d_e < 1.1 \times 10^{-29}\,e$ cm. Moreover, if we assume $\rho_{ct}$ to be small, the constraint on $\rho_{tc}$ from the charm chromo-EDM also vanishes [58]. We therefore define our specific baryogenesis scenario as [31]

$$|\rho_{tc}| \gtrsim 0.5 \qquad \text{and} \qquad |c_\gamma| \gtrsim 0.1\,, \tag{9}$$

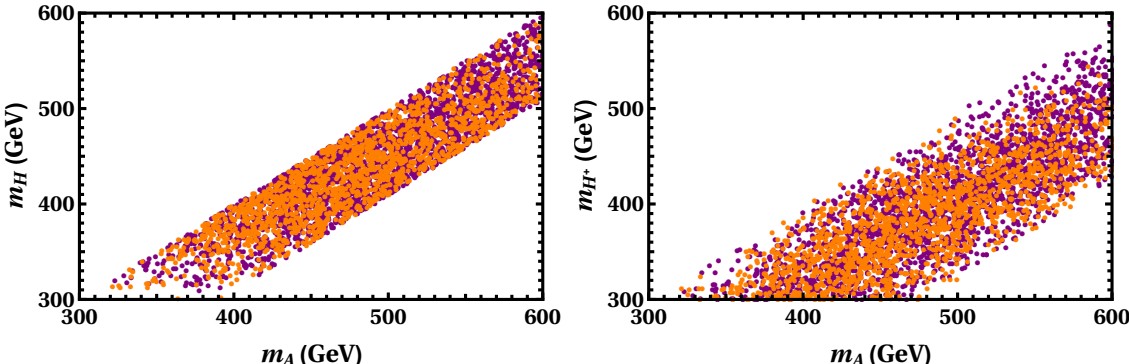

Figure 1: Parameter space allowed by perturbativity, positivity, unitarity, and electroweak precision measurements in the $m_A$–$m_H$ and $m_A$-$m_{H^\pm}$ planes. The purple and orange scanned points corresponds to $c_\gamma = 0.1$ and $0.3$ respectively.

with a sufficiently large complex phase. A strong first-order phase transition is then possible for [59–68]

$$m_{A,H,H^+} \sim 300 \ldots 600 \text{ GeV}. \tag{10}$$

This mass range is allowed by perturbativity, positivity, unitarity, and electroweak precision data. We rely on 2HDMC [69] to provide the results of Fig. 1 for $c_\gamma = 0.1$ and $0.3$. The 2HDMC parameters in the Higgs basis are $\eta_{1,\ldots,7}$ and $m_{H^\pm}$. To save computing time we actually scan $\mu_{22} \in [0, 1000]$ GeV, $m_A \in [300, 600]$ GeV, $m_H \in [300, 600]$ GeV, $m_{H^\pm} \in [300, 600]$ GeV, $\eta_2 \in [0, 6]$, and $\eta_7 \in [-6, 6]$, and express them in terms of the $\eta_i$. To match the 2HDMC conventions we define $\gamma \in [-\pi/2, \pi/2]$. We refer readers to Refs. [39, 51, 70–72] for further details on the parameter scan.

As the first constraint on $c_\gamma$ and the set of $\rho_{ij}$ we consider measurements of the SM-like Higgs. Higgs coupling measurements constrain the Higgs mixing angle to $c_\gamma \leq 0.3$ and 95%CL. Our choice of $\rho_{tc}$ as the source of $CP$-violation is motivated by its much weaker constraints, because it hardly affects SM-like Higgs production and decay. The relevant constraints on $\rho_{tc}$ are indirect. For flavor observables, $\rho_{tc}$ enters through loops with charm quarks and a charged Higgs into $B_s - \bar{B}_s$ mixing and $\mathcal{B}(B \to X_s\gamma)$. Reinterpreting the limit from Ref. [73] we find

$$|\rho_{tc}| \lesssim 1, \qquad \text{for} \quad m_{H^+} = 300 \text{ GeV}, \tag{11}$$

and its counterpart for $m_{H^+} = 500$ GeV is illustrated in Fig. 2, alongside with the EWBG-region. The limit is relatively weak in our general model, in contrast to the type-II 2HDM, and for larger $m_{H^+}$ it rapidly becomes irrelevant. Finally, finite $c_\gamma$ in combination with $\rho_{tc}$ [26] leads to anomalous top decays $t \to ch$ [28], forbidden at tree level in the SM. The current Run 2 limits at 95%CL are

$$\mathcal{B}(t \to ch) \approx \frac{c_\gamma^2 |\rho_{tc}|^2}{7.66 + c_\gamma^2 |\rho_{tc}|^2} < \begin{cases} 1.1 \times 10^{-3} & \text{ATLAS [41]} \\ 4.7 \times 10^{-3} & \text{CMS [40]}. \end{cases} \tag{12}$$

They get weaker for smaller $c_\gamma$ and vanish in the alignment limit. We illustrate the stronger ATLAS [41] constraint also in Fig. 2, along with the projected HL-LHC 95%CL upper limit $\mathcal{B}(t \to ch) < 1.0 \times 10^{-4}$ [74]. While an observation of this anomalous decay could point to a large value of $|\rho_{tc}|$, if would not provide a link to baryogenesis. A natural step towards solving the baryogenesis puzzle would be to search for new scalar degrees of freedom related to this flavor-changing coupling.

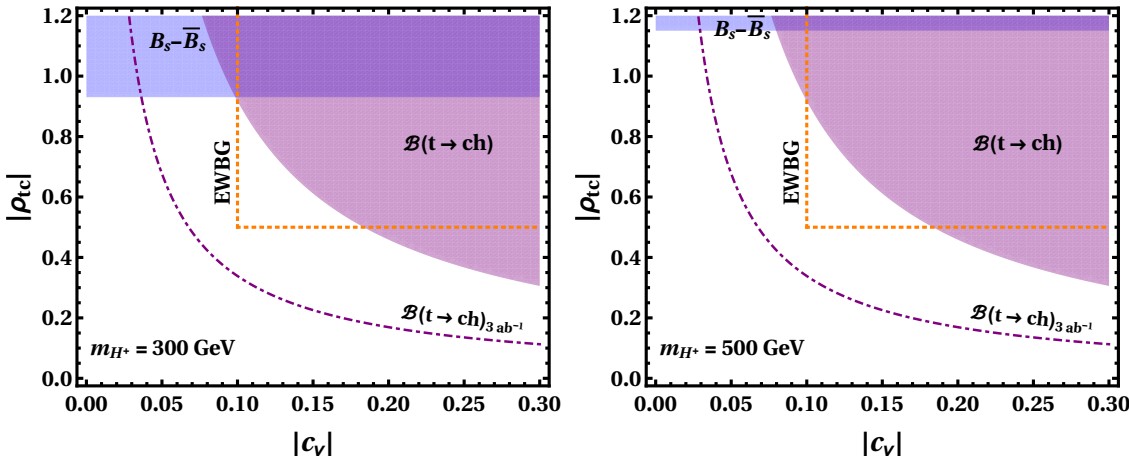

Figure 2: Indirect constraints from $B_s - \overline{B}_s$ mixing (blue), $\mathcal{B}(t \to ch)$ (purple) in the $\rho_{tc}$–$c_\gamma$ plane for two $H^+$ mass values, together with the baryogenesis region (orange).

While we will focus on $\rho_{tc}$ throughout this paper, we point out that $\rho_{tu}$ can be tested using a very similar strategy. For the LHC processes discussed in the coming sections, there is always a corresponding process with an up-quark replacing the charm-quark. One difference between the two FCNC scenarios is that $\rho_{tu}$ can induce observable effects in $\mathcal{B}(B \to \mu\nu)$ [75], within the reach of Belle-II [76]. The combination of $\rho_{tc}$ and $\rho_{tu}$ is subject to very strong constraints from $D$–$\overline{D}$ mixing [73], and we will assume only one of the two, but not both at the same time.

## 3 Charged Higgs production

In the EWBG parameter region of Eq.(9), the partonic process at LHC

$$cg \to bH^+ \to b\,(W_\ell^+ h) \to b\,W_\ell^+ W_\ell^+ W_\ell^-, \tag{13}$$

probes $\rho_{tc}$ in $H^+$-production and $c_\gamma$ in the decay $H^+ \to W^+ h$. The production benefits from the relatively large charm density in the proton, as well as the combination [39] with the CKM matrix element $V_{tb}$ following Eq.(7). The leading-order Feynman diagrams are presented in Fig. 3. While we will require a tagged $b$-jet, the $b$-inclusive production process could also be defined as $c\bar{b} \to H^+$ [77,78]. For a clean analysis, we assume that all three $W$-bosons decay to either electrons or muons. The same process is induced by $\rho_{ct}$, but this coupling is constrained to be much smaller [79] by flavor constraints.

Table 1: Charged Higgs properties for the two benchmark points with $\rho_{tc} = 0.35$ and $c_\gamma = 0.25$. The quoted LHC cross sections include the decay $H^+ \to Wh$ in the fully leptonic mode, as shown in Eq.(13), as well as selection and background rejection cuts.

| $m_{H^+}$[GeV] | $\Gamma_{H^+}$[GeV] | $\mathcal{B}(H^+ \to c\bar{b})$ | $\mathcal{B}(H^+ \to W^+ h)$ | $\sigma(cg \to bH^+)$[fb] |
|---|---|---|---|---|
| 350 | 2.2 | 0.85 | 0.15 | 0.126 |
| 500 | 3.9 | 0.66 | 0.34 | 0.113 |

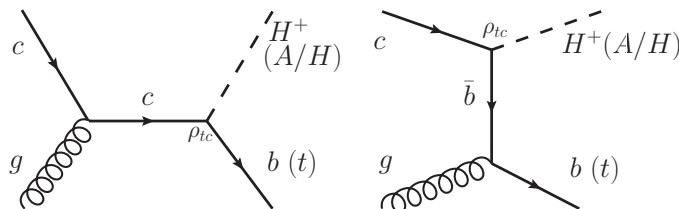

Figure 3: Leading-order Feynman diagrams for the $\rho_{tc}$-induced $cg \to bH^+$ and $cg \to tA/tH$ processes.

The $H^+W^-h$ coupling, modulated by $c_\gamma$, arises from [13,14]

$$\mathcal{L} \supset -\frac{g_2}{2}c_\gamma \left(h\partial^\mu H^+ - H^+\partial^\mu h\right) W_\mu^- + \text{h.c.}, \tag{14}$$

where $g_2$ is the $SU(2)$ gauge coupling. To estimate the reach of our charged Higgs signal, we choose two allowed benchmark points,

$$\rho_{tc} = 0.35, \qquad c_\gamma = 0.25, \qquad m_{H^+} = 350, 500 \text{ GeV}, \tag{15}$$

as given in Tab. 1. For the branching ratios, we ignore the loop-induced decays $H^+ \to W^+\gamma$ and $H^+ \to W^+Z$. We generate signal and background events for $\sqrt{s} = 14$ TeV at leading order with MadGraph5_aMC@NLO [80]. The effective model is implemented in the FeynRules [81] framework, and for parton densities we use NN23LO1 [82]. The events are showered and hadronized with PYTHIA6.4 [83] and then handed to Delphes 3.4.2 [84] for a fast detector simulation with the default ATLAS card. Jets are reconstructed with an $R = 0.6$ anti-$k_T$ algorithm [85] in FastJet [86]. For $b$-tagging as well as $c$-jet and light-jet rejections, we also rely on the default ATLAS card. To allow for extra jets we apply MLM matching [87,88] with the default MadGraph5_aMC run card. The signal is generated with up to two additional jets, do account for higher-order effects in the event kinematics.

The dominant SM-backgrounds are $t\bar{t}W$ and $t\bar{t}Z$ production, followed by $WZ +$ jets, $4t$, $t\bar{t}h$, $tZj$, $tWZ$, and $ZZ +$ jets. Furthermore, we find the backgrounds $3t$, $3t +W$, and $3W$ to be negligible, so we ignore them in our analysis. However, given a mis-identification probability for a jet as a lepton around $10^{-4}$ [89,90], $t\bar{t}$ production will lead to non-trivial background contributions. For all backgrounds, we use the same simulation chain as for the signal, with up to one additional jet for $t\bar{t}W$, $t\bar{t}Z$, $WZ +$ jets, $ZZ +$ jets, $tZ +$ jets, $t\bar{t}+$ jets, and no QCD jets for the high-multiplicity backgrounds $4t$, $tWZ$ and $t\bar{t}h$. To approximately account for QCD corrections in addition to the jet emission, we attach NLO $K$-factors to the dominant $t\bar{t}V$ backgrounds, namely 1.35 ($W^-$), 1.27 ($W^+$) [91], and 1.56 ($Z$) [92]. We also correct the $WZ +$ jets and $t\bar{t}+$ jets background normalizations to NNLO by factors 2.07 [93] and 1.84 [94] respectively. Furthermore, we adjust the $4t$, $t\bar{t}h$, and $\bar{t}Z +$ jets rates to NLO through the $K$-factors 2.04 [80], 1.27 [95] and 1.44 [80]. The cross sections for the signal and $tWZ$ are kept at LO for simplicity. Here, we simply assume the QCD correction factors for the $W^+Z +$ jets and $tZ +$ jets processes to be the same as their respective charge-conjugate processes.

To suppress the backgrounds, we adopt a simple set of requirements. We start with events containing at least three charged leptons and at least one tagged $b$-jet passing

$$\begin{aligned}
p_{T,\ell} &> 20 \text{ GeV}, & |\eta_\ell| &< 2.5, & \\
p_{T,b} &> 20 \text{ GeV}, & |\eta_b| &< 2.5, & \\
\Delta R_{ij} &> 0.4, & &- & (i,j = \ell, b), \\
\not{E}_T &> 35 \text{ GeV}, & m_{\ell^+\ell^-} &\notin [76, 110] \text{ GeV}, & (\ell = e, \mu).
\end{aligned} \tag{16}$$

Table 2: Background cross sections for the charged Higgs process after cuts.

| | $t\bar{t}W$ | $tt\bar{Z}$ | $WZ$+jets | $4t$ | $t\bar{t}h$ | $tZ$+jets | $tWZ$ | $ZZ$+jets | $t\bar{t}$+jets | sum bkg |
|---|---|---|---|---|---|---|---|---|---|---|
| merged jets | 1 | 1 | 1 | 0 | 0 | 1 | 0 | 1 | 1 | |
| $K$-factor | NLO | NLO | NNLO | NLO | NLO | NLO | LO | LO | NNLO | |
| $\sigma_{\text{bkg}}$ [fb] | 0.685 | 0.279 | 0.101 | 0.074 | 0.026 | 0.017 | 0.02 | 0.001 | 0.304 | 1.504 |

The same-flavor opposite-sign dilepton veto reduces the dominant $t\bar{t}Z$ background. In case more than one such $\ell^+\ell^-$ pair exists, we select the combination closest to the $Z$-mass for rejection. The remaining signal rate is given in Tab. 1, while the background rates are summarized in Tab. 2.

For discovery reach and exclusion limits, we compute the significance using the likelihood for a simple counting experiment [96]. If we observe $n$ events with $n_{\text{pred}}$ predicted, the agreement between observation and prediction is given by

$$Z(n|n_{\text{pred}}) = \sqrt{-2\ln\frac{L(n|n_{\text{pred}})}{L(n|n)}}, \qquad \text{with} \qquad L(n|\bar{n}) = \frac{e^{-\bar{n}}\bar{n}^n}{n!}. \tag{17}$$

For discovery, we compare the observed signal plus background with the background prediction and require $Z(s+b|b) > 5$. For exclusion, we assume a background-consistent measurement after predicting a signal on top of the background, such that $Z(b|s+b) > 2$. For instance, assuming an HL-LHC data set with 3000 fb$^{-1}$ and the signal and background cross sections in Tabs. 1 and 2, we find a significance of $\sim 5.6\sigma$ for $m_{H^+} = 350$ GeV and $\sim 5\sigma$ for $m_{H^+} = 500$ GeV.

We illustrate in Fig. 4 the Run 3 and HL-LHC reach for the charged Higgs signature in the $|\rho_{tc}|$–$c_\gamma$ plane. We see from the left panel that Run 3 can exclude $|\rho_{tc}| > 0.3$ and $|c_\gamma| = 0.27$ for $m_{H^\pm} = 350$ GeV, while the HL-LHC will be sensitive to $|\rho_{tc}| > 0.2$ and $|c_\gamma| = 0.14$. For larger Higgs masses, the expected limits become only slightly weaker. The $b$-associated charged Higgs channel covers the $|\rho_{tc}|$ range preferred by EWBG, but there remains a slice of EWBG parameter space with $|c_\gamma| \lesssim 0.14$. This follows as an effect of decreasing $\mathcal{B}(H^\pm \to W^\pm h)$ with smaller $c_\gamma$. Unfortunately, this hole is unlikely to be filled by other charged Higgs decays, because for instance the standard signature $H^+ \to t\bar{b}$ requires large production rates. Here, utilizing the expression from Ref. [73], the limit from $B_s$ in the left panel of Fig. 4 is plotted for $m_{H^+} = 350$ GeV to conform with the benchmark charged Higgs mass for $pp \to bH^+ \to bW^+h$ signature.

## 4 Neutral Higgs production

To cover the parameter region $|c_\gamma| < 0.14$, left open by the charged Higgs signature, we turn to the neutral Higgs channel,

$$cg \to tH/tA \to t\,(t\bar{c}), \tag{18}$$

also given in Fig. 3, where production and decay are both mediated by $\rho_{tc}$. A very slight $c_\gamma$-dependence of the $cg \to tH/tA \to tt\bar{c}$ process arises from the heavy Higgs branching ratios. Non-resonant and $t$-channel diagrams with $H/A$ exchange leading to $cc \to tt$ scattering as well as $gg \to tt\bar{c}\bar{c}$, though small, are included in our signal analysis.

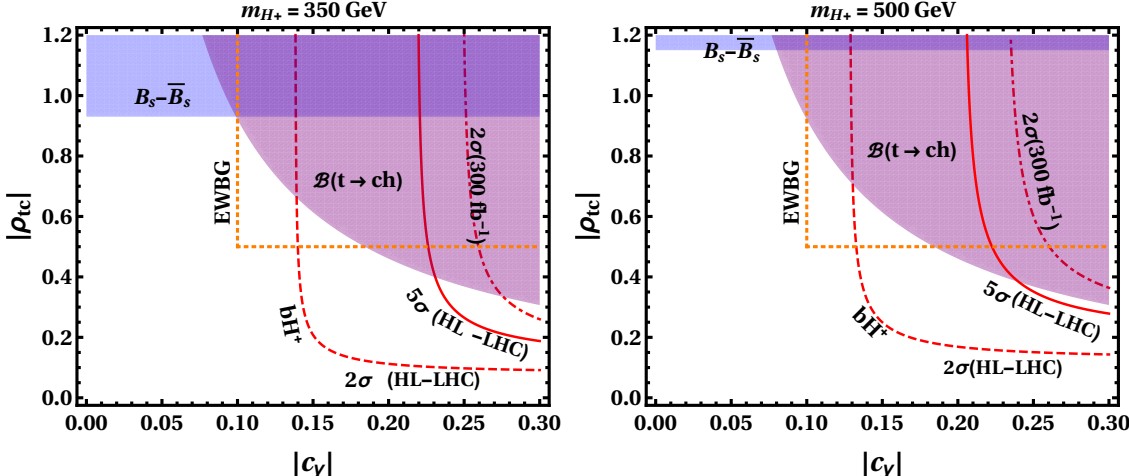

Figure 4: Projected 300 fb$^{-1}$ exclusion (dot-dashed) and HL-LHC discovery (solid) and exclusion (dashed) contours for the charged Higgs signature $pp \to bH^+ \to bW^+h$, along with EWBG-favored region and the indirect constraints from Fig. 2.

For small $c_\gamma$, the neutral Higgs production process currently leads to the most stringent limit on $\rho_{tc}$ [33, 97], because it affects the SM control region of the Run 2 $t\bar{t}t\bar{t}$ (4t) analysis by CMS [98]. Based on the number of $b$-jets and leptons, CMS divides its analysis into several signal and two control regions. The most stringent constraint on $\rho_{tc}$ arises from the $t\bar{t}W$ control region (CRW) [32, 33]. The CMS baseline selection includes two same-sign leptons with

$$p_{T,\ell} > 25, 20 \text{ GeV} \qquad \text{and} \qquad |\eta_e| < 2.5, \qquad |\eta_\mu| < 2.4, \tag{19}$$

where the charge-misidentified Drell-Yan background is reduced by vetoing same-sign electron pairs with $m_{ee} < 12$ GeV. The CRW then requires two to five jets, two of them $b$-tagged. All jets have to fulfill $|\eta_j| < 2.4$, and events are selected if they fulfill any one of

| | | | |
|---|---|---|---|
| (i) | $p_{T,b_1} > 40$ GeV, | $p_{T,b_2} > 40$ GeV, | — |
| (ii) | $p_{T,b_1} > 20$ GeV, | $p_{T,b_2} = 20 \ldots 40$ GeV, | $p_{T,j_3} > 40$ GeV, |
| (iii) | $p_{T,b_{1,2}} = 20 \ldots 40$ GeV, | — | $p_{T,j_{3,4}} > 40$ GeV. |

(20)

Finally, the analysis requires [98]

$$H_T = \sum_{\text{jets}} p_{T,j} > 300 \text{ GeV} \qquad \text{and} \qquad \not{E}_T > 50 \text{ GeV}. \tag{21}$$

With this selection, CMS observes 338 events with $335 \pm 18$ events expected from SM-backgrounds plus 4t signal. To estimate the CRW limits on $\rho_{tc}$, we generate both neutral Higgs processes with the decay $H/A \to t\bar{c}$, followed by lepton-hadron combinations of the top decays at $\sqrt{s} = 13$ TeV. We use the same setup as for the charged Higgs simulations, except that we use the default CMS detector card in Delphes 3.4.2. Remaining uncertainties on our simulation affect the $c$-initiated processes $cg \to bH^+$ and $cg \to tA/tH$, such as from parton densities and scale dependence [78, 99–101]. We expect them to be small, and do not include them, just as we do not account for non-prompt and fake backgrounds.

There exist a similar ATLAS search [102], but it is less constraining [103]. This is primarily due to the definition of signal regions and selection criteria. Furthermore, searches for squark

Table 3: Background cross sections for the dedicated same-sign top search after selection cuts at $\sqrt{s} = 14$ TeV.

| background | $\sigma$ [fb] | background | $\sigma$ [fb] | background | $\sigma$ [fb] |
|---|---|---|---|---|---|
| $t\bar{t}W$ | 1.31 | $t\bar{t}Z$ | 1.97 | $tZ + \text{jets}$ | 0.007 |
| $4t$ | 0.092 | $3t + W$ | 0.001 | $3t + j$ | 0.0004 |
| $t\bar{t}h$ | 0.058 | charge-flip | 0.024 | non-prompt | $1.5\times\, t\bar{t}W$ |

pair production in $R$-parity violating supersymmetry [104] and exotics searches for same-sign dileptons and $b$-jets [105] involve similar final states, but their selection cuts are too model-specific to be applied to our signature.

To judge the impact of the existing CMS CRW limits from $4t$ search, we focus on the border of the EWBG-region with $c_\gamma = 0.1$ and $|\rho_{tc}| = 0.5$. We stick to our two charged Higgs masses, assume $m_A \approx m_{H^\pm} = 350, 500$ GeV for the pseudoscalar, and decouple the heavy scalar $H$. In this scenario, the same-sign top contribution to the CRW arises from $cg \to tA \to tt\bar{c}$. We demand that the combination of SM-backgrounds and heavy neutral Higgs production agree with observed within $2\sigma$ and give the excluded regions in Fig. 5. To scan the parameter space we use a simplified scaling $|\rho_{tc}|^2 \mathcal{B}(A \to t\bar{c})$, such that $\Gamma_A = 3.05(6.08)$ GeV for $m_A = 350(500)$ GeV. The exclusion covers most of the EWBG-region except for small values of $|\rho_{tc}|$.

A dedicated same-sign top search, such as the $pp \to tA + X \to tt\bar{c} + X$ study of Ref. [103], can probe the nominal parameter space of $\rho_{tc}$-EWBG. This process can be searched for in events containing same-sign dileptons ($ee$, $\mu\mu$, $e\mu$), at least three jets with at least two $b$-tag, and some $\not{E}_T$. The dominant backgrounds are $t\bar{t}Z$, $t\bar{t}W$, $4t$, while $t\bar{t}h$, with $tZ + \text{jets}$, $3t + W$ and $3t + j$ give subdominant contributions, and the non-prompt background can be 1.5 times the rate of $t\bar{t}W$. In addition, if a lepton charge gets misidentified, the $t\bar{t} + \text{jets}$ and $Z/\gamma^* + \text{jets}$ processes will also contribute. For further details of the QCD correction factors for different backgrounds, we refer to Ref. [103]. To reduce backgrounds, we applied an event selection different from the CRW of Ref. [98]: the leading and sub-leading same-sign leptons should have $p_T > 25(20)$ GeV and $|\eta| < 2.5$. All three jets are required to have $p_T > 20$ GeV and $|\eta| < 2.5$. All jets and leptons are separated by $\Delta R_{ij} > 0.4$. The all event should have $\not{E}_T > 35$ GeV. and $H_T > 300$ GeV, where the latter includes the two leading sames-sign leptons. The background cross sections after selection cuts are summarized in Tab. 3.

For the reference values $|\rho_{tc}| = 0.5$ and $c_\gamma = 0.1$, we generate the same-sign top cross sections for $m_A = 350$ and $500$ GeV. Based on the background rates of Tab. 3 and Eq.(17), rescaling the signal cross section by $|\rho_{tc}|^2 \mathcal{B}(A \to t\bar{c})$, we find the exclusion (green dashed) and discovery (green solid) contours in the $|c_\gamma|$–$|\rho_{tc}|$ plane as given in Fig. 5.

A loop hole in the neutral Higgs analysis appears though the destructive interference of $cg \to tH \to tt\bar{c}$ and $cg \to tA \to tt\bar{c}$. If the widths and masses of the two heavy neutral Higgses become degenerate, the two production processes completely cancel [32, 33] and the same-sign top signature vanishes. Our limits derived from $A$-production would be similar for $H$-production with $m_A \gg m_H$. We now illustrate limits for $m_A \sim m_H$ with a case where the three heavy Higgs masses are of similar size, specifically $m_{H^\pm} = 350(500)$ GeV, $m_A = 343(524)$ GeV, and $m_H = 355(501)$ GeV. The self-couplings are $\eta_1 = 0.276(0.297)$, $\eta_2 = 1.335(2.762)$, $\eta_3 = 1.66(1.21)$, $\eta_4 = -0.04(0.398)$, $\eta_5 = 0.121(-0.428)$, $\eta_6 = -0.181(-0.386)$, $\eta_7 = 0.605(-0.095)$, and $\mu_{22}^2/v^2 = 1.189(3.516)$, in agreement with perturbativity, positivity, unitarity, and electroweak precision data [69]. The relevant decays are $A \to t\bar{c}, Zh$ and $H \to t\bar{c}, hh, ZZ, WW$, with mild contributions from the $\lambda_f c_\gamma$-dependent fermionic decays to $b\bar{b}$ and $t\bar{t}$. For $\rho_{tc} = 0.5$ and $c_\gamma = 0.1$, the total widths are $\Gamma_A = 3.28(7.37)$ GeV and

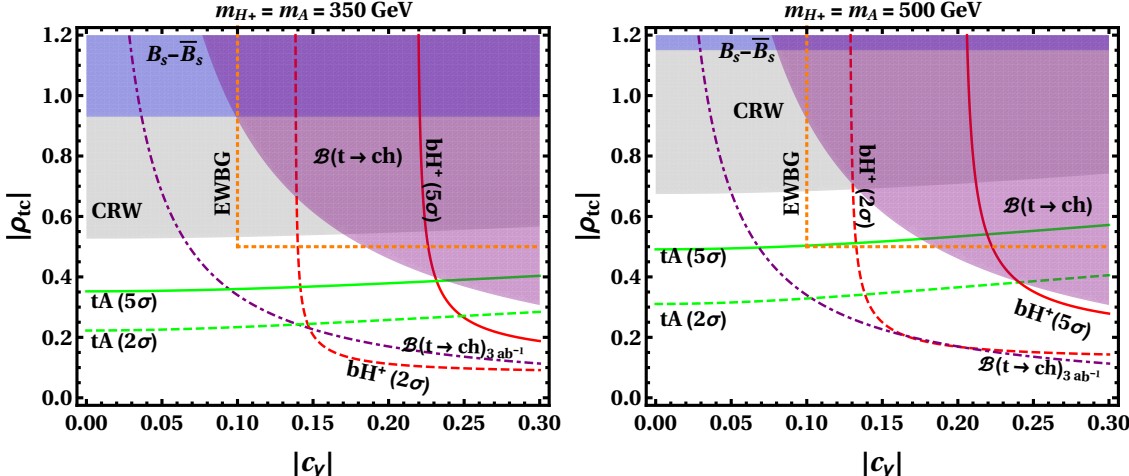

Figure 5: Exclusion regions from neutral Higgs production contributing to the CMS CRW [98] (gray shades), as well as HL-LHC expectations from a dedicated same-sign top search [103] (green). We also show the EWBG region and the indirect constraints from Fig. 2 and the HL-LHC charged Higgs reach from Fig. 4.

$\Gamma_H = 2.91\,(6.56)$ GeV, and the combined contributions to the CRW rates are 0.467 fb and 0.261 fb, corresponding to 64 and 35.8 events. Demanding that the combination of events expected in the SM and from the neutral Higgs channels agree within $2\sigma$ of the observed number, we find that $|\rho_{tc}| = 0.5$ is already excluded for $m_{H^\pm} = 350$ GeV and $c_\gamma = 0.1$, and barely allowed for $m_{H^\pm} = 500$ GeV. We see that, due to the choice of parameters, the cancellation between $cg \to tH \to tt\bar{c}$ and $cg \to tA \to tt\bar{c}$ is not exact, and the CRW limit is stronger than the $H$ (or $A$) decoupled case.

As mentioned in the introduction, we ignore all $\rho_{ij}$ couplings except for $\rho_{tc}$, so before closing we should discuss the impact of this assumption. It may well be that the $\rho^{U,D,L}$ matrices share the flavor-ordering of the Yukawa couplings, $\rho_{tt} \sim \lambda_t$, $\rho_{bb} \sim \lambda_b$ and $\rho_{\tau\tau} \sim \lambda_\tau$. Current data still allows $\rho_{tt} \lesssim 0.5$ [39] and $\rho_{bb} \sim 0.1$ [71,72] for sub-TeV scalars, and both parameters can account for the observed baryon asymmetry. The extra top Yukawa coupling $\rho_{tt}$ can be searched for in signatures such as $gg \to A/H \to t\bar{t}$ [106, 107] $gg \to A/Ht\bar{t} \to t\bar{t}t\bar{t}$ [98] and $gb \to \bar{t}H^+ \to t\bar{t}\bar{b}$ [77, 78, 108, 109], while rare decays $\mathcal{B}(B \to X_s\gamma)$ and $B_{d,s}$ mixing provide indirect probes [79]. In general, a large value for $\rho_{tt}$ dilutes the decays $A/H \to t\bar{c}$ and $H^+ \to W^+h$ through $A/H \to t\bar{t}$ and $H^+ \to t\bar{b}$. However, the combination with $\rho_{tc}$ opens additional discovery modes such as $cg \to tA/tH \to tt\bar{t}$ [32] and $cg \to bH^+ \to bt\bar{b}$ [39]. There also exist several direct and indirect constraints on $\rho_{bb}$ [71,72]. Finally, a large allowed value of $\rho_{tu}$ [103] combined with non-vanishing $\rho_{tc}$ will be constrained by $D$-meson mixing [73, 79]. Similarly, constraints on $\rho_{tt}$, $\rho_{bb}$, $\rho_{\tau\tau}$ from flavor physics and low energy observables as discussed in Refs. [38,72,73,79], and their detailed impact on the $\rho_{tc}$-EWBG would be an interesting future direction.

## 5 Outlook

Electroweak baryogenesis is an attractive target for experimental analysis, because it can be tested by a variety of measurements. Specific models typically combine new bosonic degrees of freedom with extra $CP$-violation. In our case, the new degrees of freedom are provided by

a general or type-III 2HDM. If the Higgs self-couplings are sufficiently large, the heavy Higgs states can be relatively heavy, so we use $m_{H^+} = 350$ and $500$ GeV as benchmark scenarios. The complex phase is given by an FCNC top–charm coupling with $|\rho_{tc}| \gtrsim 0.5$, combined with a $CP$-even Higgs mixing angle $c_\gamma \gtrsim 0.1$. At the LHC, $\rho_{tc}$ has the advantage that we can test it in processes mediated by this large top Yukawa, but with a charm quark in the initial state, while it easily evades EDM constraints.

In the allowed 2HDM parameter space, the charged Higgs has to be relatively light, which means we can search for it via $cg \to bH^+$ with a subsequent $H^+ \to W^+ h$ decay. Our proposed analysis is relatively straightforward and probes most of the EWBG parameter space at the HL-LHC, with the exception of small values of $c_\gamma \sim 0.1 \dots 0.12$, when $H^+ \to W^+ h$ decay becomes too suppressed by $CP$-even Higgs boson mixing.

A complementary channel that can survive small $CP$-even Higgs boson mixing is heavy neutral Higgs production, $cg \to tA/tH$, together with $A/H \to t\bar{c}$ decay. In this case, production and decay are both mediated by $\rho_{tc}$ without being suppressed by small $c_\gamma$, providing strong limits on $\rho_{tc}$ even for small $c_\gamma$ values. The search channel at the LHC is same-sign top pairs, allowing us to extract limits already from Run 2. At the HL-LHC, the decay $t \to ch$, charged heavy Higgs searches, and neutral heavy Higgs searches guarantee a comprehensive coverage of the $\rho_{tc}$-EWBG parameter space in the general 2HDM, leaving us with the challenge of observing the $CP$-violating phase in a dedicated analysis.

# Acknowledgments

First, we would like to thank Kai-Feng (Jack) Chen for discussions and for clarifications on Eq.(20). We are also grateful to Eibun Senaha and Margarete Mühlleitner for very helpful discussions and comments. WSH is supported by MOST 109-2112-M-002-015-MY3 of Taiwan and NTU 109L104019. TM is supported by a Postdoctoral Research Fellowship from Alexander von Humboldt Foundation. The research of TP is supported by the Deutsche Forschungsgemeinschaft (German Research Foundation) under grant 396021762 – TRR 257 *Particle Physics Phenomenology after the Higgs Discovery*.

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
