# Peer review of "A Final Word on FCNC-Baryogenesis from Two Higgs Doublets"

_SciPost Physics, doi:SciPost Phys. 10, 150 (2021)_

## Round 1 · Referee Report · Anonymous (Referee 1) · 2021-1-15

Report

The authors study the possibility of BG in a general THDM. They conclude that with future LHC Run 3 or HL-LHC searches this scenario can fully be tested. I cannot recommend the paper for publication, since there are several open and unresolved issues. These are the following:

model definition:

The potential is defined in eq. (4) and the couplings to fermions in eq. (6). It remains unclear what are the input parameters of the model. The various choices of the rho matrices have to be deduced from information scattered over the article. The choide of the elements of the rho matrices appears fully ad-hoc to satisfy experimental constraints. Is there any underlying model that would produce such a extremely weird pattern that the authors employ?

checks on the model parameters:

There is hardly any information on the checks performed. The authors mention in passing perturbativity, positivity, unitarity, and electroweak precision data. It remains completely unclear what the effects of these are on the model as such and on the particular parameter choices the authors make. Completely absent seem to be checks for the properties of the SM-like Higgs boson at 125 GeV, as well as checks for the BSM Higgs searches for the additional Higgs bosons (except the particular channels the authors are analysing later).

Fig. 1:

What is the point in showing it for MH+- = 300 GeV and 500 GeV? The only difference seems to come from Bs-Bsbar mixing. What are the other relevant parameters here? How do they influence the excluded regions? Furthermore, it appears that the searches for t -> ch can rule out the scenario under investigation in the (near?) future. The authors do not comment on this.

"usual" BSM Higgs production and decay modes:

The authors are investigating solely the channels they are interested in. However, there should be some "usual" channels, such as gb -> tH+-, H+- -> tb, which may be suppressed (because of the weird and unmotivated choice of rho^U,D), but via CKM mixing they should still play a significant role. The authors do not comment on this.

Calculation of signal and background:

Some channels are calculated at NLO, others at LO, which in particular seems to include the signal channels. Why? I do not see any motivation for this from the physics point of view.

Fig. 4:

It seems that what the authors label "CRW" alone is already excluding a lot of the allowed parameter space, so this should be looked at first, since it is an existing limit. Also this channel, which is not in the main focus of the authors, seems to be able to rule out their scenario with future LHC runs. Again this is not discussed.

I do not include several smaller issues that I have, e.g. concerning citations, since the above issues are far more relevant. But looking at Ref. [84] I wonder whether the authors really read their article before submitting it.

  • validity: low
  • significance: low
  • originality: ok
  • clarity: low
  • formatting: -
  • grammar: -

Author:  Tanmoy Modak  on 2021-02-25  [id 1267]

(in reply to Report 1 on 2021-01-15)

model definition:

The potential is defined in eq. (4) and the couplings to fermions in eq. (6). It remains unclear what are the input parameters of the model. The various choices of the rho matrices have to be deduced from information scattered over the article. The choide of the elements of the rho matrices appears fully ad-hoc to satisfy experimental constraints. Is there any underlying model that would produce such a extremely weird pattern that the authors employ?

Our primary focus is to probe the parameter space required for rho_tc-EWBG. For practical purposes, we have turned off all rho_ij couplings except for rho_tc. We have added a discussion in the last paragraph of Sec.4 on the impact of turning on other rho_ij couplings. The dynamical parameters in the 2HDM potential would be subjected to constraints from perturbativity, positivity, unitarity, and oblique parameters. We have now added discussion regarding this in the first paragraph of page 4.

checks on the model parameters:

There is hardly any information on the checks performed. The authors mention in passing perturbativity, positivity, unitarity, and electroweak precision data. It remains completely unclear what the effects of these are on the model as such and on the particular parameter choices the authors make. Completely absent seem to be checks for the properties of the SM-like Higgs boson at 125 GeV, as well as checks for the BSM Higgs searches for the additional Higgs bosons (except the particular channels the authors are analysing later).

We have added discussions and parameter scans (see Fig.1) for the allowed parameter space by perturbativity, positivity, unitarity, and electroweak precision data (checked via 2HDMC). As we turn off all rho_ij couplings, and given that rho_tc does not enter the Higgs signal strengths at tree level, the impact is minimal. However the mixing angle c_gamma would receive some constraints via hZZ, hWW etc. vertices. We had already checked that the chosen c_gamma range [0.1,0.3] is allowed by Higgs STXS measurements. We now added discussion in second paragraph of page 4.

Fig. 1:

What is the point in showing it for MH+- = 300 GeV and 500 GeV? The only difference seems to come from Bs-Bsbar mixing. What are the other relevant parameters here? How do they influence the excluded regions? Furthermore, it appears that the searches for t -> ch can rule out the scenario under investigation in the (near?) future. The authors do not comment on this.

The plots for mH+ = 300 and 500 GeV are chosen for illustration. Along with the B_s mixing, the reach of cg > b H+ > b h W+ process is also different for the respective masses. In the previous version we inadvertently put the wrong 2sigma contour for the HL-LHC exclusion from cg > b H+ > b h W+ process in the left panel of Fig.4 (previous version). Please see the modified left panel Fig.5 in the latest version for comparision. We have added the projected HL-LHC reach for the t > ch decay, including comments in the first paragraph of page 5.

"usual" BSM Higgs production and decay modes:

The authors are investigating solely the channels they are interested in. However, there should be some "usual" channels, such as gb -> tH+-, H+- -> tb, which may be suppressed (because of the weird and unmotivated choice of rho^U,D), but via CKM mixing they should still play a significant role. The authors do not comment on this.

Indeed, due to our choice of parameters such processes are not relevant and we have checked the constraints from searches such as gb -> tH+-, H+- -> tb are well within the experimental reach with our choice of parameters. We added discussions regarding this in the last paragraph of Sec 4.

Calculation of signal and background:

Some channels are calculated at NLO, others at LO, which in particular seems to include the signal channels. Why? I do not see any motivation for this from the physics point of view.

For the charged Higgs production cg > b H+ > b h W+ we have normalised our LO background cross sections upto NLO (at least) for all backgrounds, except for the SM tWZ background which is subsubdominant. Same is true for neutral Higgs production cg > tA/tH > t t cbar. In addition, we have considered at least one additional jets in the final state for all dominant backgrounds and signals. However, a full estimation of QCD corrections for the signal processes is beyond the scope of the current paper. Therefore we believe our discovery and exclusion contours are conservative.

Fig. 4:

It seems that what the authors label "CRW" alone is already excluding a lot of the allowed parameter space, so this should be looked at first, since it is an existing limit. Also this channel, which is not in the main focus of the authors, seems to be able to rule out their scenario with future LHC runs. Again this is not discussed.

The cg > tA/tH > t t cbar process can indeed probe a large part of the parameter space for |rho_tc|. However, we re-iteriate that the c_gamma dependence is very mild in cg > tA/tH > t t cbar, as can be seen from Fig.5. It is solely for this reason we put discussion on cg > b H+ > b h W+ process prior to neutral Higgs production cg > tA/tH > t t cbar. We have elevated the discussion in the appendix and merged it into the main text. As mentioned in the previous version (see also reply for Referee 2) the CRW region of Ref.[100] is not optimized for the cg > tA/tH > t t cbar process. Therefore we advocate a simple dedicated search for cg > tA/tH > t t cbar process as discussed in Sec.4.

I do not include several smaller issues that I have, e.g. concerning citations, since the above issues are far more relevant. But looking at Ref. [84] I wonder whether the authors really read their article before submitting it.

We use its estimate of the NLO SM ttbarh cross section to determine the QCD correction factor for SM ttbarh background. We are not entirely sure what the problem is and why this has to be phrased in a rather unprofessional and not even funny manner.

---

## Round 1 · Referee Report · Anonymous (Referee 2) · 2021-1-17

Report

The present study investigates LHC constraints on the baryogenesis-motivated parameter region of a type-III two-Higgs-doublet model, where the necessary CP violation arises through a complex flavour-changing Higgs-top-charm coupling rho_tc. It is a follow-up on a series of papers by one of the authors on this model, including Ref. 25, which studied the aspect of electroweak baryogenesis, and a recent PRL article (Ref. 28), where the collider signatures in presence of non-zero rho_tc were elaborated.

Indeed, while not explicetly focussing on the baryogenesis-motivated parameter region, Ref. 28 already presents the most important constrains from searches for heavy charged and neutral Higgs states. In particular the it is already pointed out that the ttW control region (CRW) of the CMS 4top analysis with 137/fb excludes rho_tc >~0.4. A follow-up paper on the short PRL article with the technical details of the collider analyses would in principle be interesting, if done properly. The present manuscript attempts this for the specific case of small rho_tt but sizeable rho_tc, but unfortunately falls short of the necessary details and scrutiny.

  • Introduction and section 2:

  • As also critisized by the first referee, the model is not motivated well in the paper, nor is the considered parameter space. The reader is left to search the previous literature for details. The relevant references are mentioned throughout the text, but no consistent overview of the previous literature leading up to this work is given. The couplings lambda_ii and rho_ij are introduced completely ad hoc in the introduction; while some explanation comes in section 2, one has to make an effort to piece things together while reading.

  • Refs [3-5] are an extremely selective list of theorists' fits of the Higgs couplings. There exists a vast literature on this subject, most of which is ignored here. Moreover, and perhaps more importantly, dedicated measurements of the Higgs couplings have been published directly by ATLAS and CMS.

  • Eq. (1) is duplicated in eq. (8).

  • It is not clear what are the actual constraints on rho_tt, and what are the consequences for this study if both rho_tt and rho_tc are relevant.

  • Section 3

  • "For b-tagging as well as c-jet and light-jet rejection, we rely on Delphes" is too vague. The standard ATLAS card can change over time, so details on the assumed reconstruction efficiencies must be given, and not only for (b)jets but also for lepton ID. The same is true for extra jets: details on the matching and merging parameters are necessary.

  • Cross sections in Table 2: while it can be justified to compute sub-leading contributions at lower order, this needs to be justified and explained properly. "for simplicity" is not good enough.

  • How many events were actually generated?

  • Run 3 and HL-LHC reach: to what extent are the efficiencies of the standard ATLAS Delphes card appropriate for HL-LHC simulation?

  • What about pile-up?

  • Section 4

  • Same concerns as above for standard CMS Delphes card and details on the event generation.

  • The CMS 4 top analysis has been reinterpreted with MadAnalysis5 in Phys.Lett.B 784 (2018) 223 [1805.10835]. There, a specific Delphes card was developed to reproduce well the CMS results. This was for 35/fb of integrated luminosity, but an update of the recast code (and appropriately tuned Delphes card) are available also for the 137/fb version, see https://madanalysis.irmp.ucl.ac.be/wiki/PublicAnalysisDatabase While these recast codes consider only the signal regions of the CMS analysis but not the control regions, the fact that a specific tuning was necessary raises severe doubts about the validity of using the standard CMS Delphes card in the present work. How has this been validated? Details on this must be explained and the accuracy of the reinterpretation demonstrated.

  • The same-sign top search is a crucial for this work. Why is it relegated to the appendix? Again, relevant details are not explained.

  • Given the very strong conclusions about ruling out the model under investigation, the impact of the uncertainties mentioned in the last paragraph before the conclusions should be quantified.

In conclusion, while the study is interesting in principle, much more detail (and care) is needed to make the paper suitable for publication. I actually have more comments on the paper, but given the shortcomings listed above I refrain from going into finer details in this iteration.

  • General remark: for the sake of reproducibility, the MadGraph model (UFO) file, all the Monte Carlo settings for the event generation (i.e. the MG5, Pythia, Delphes cards) and the analysis codes should be made public on an appropriate versioned and citable open-access repository, for example Zenodo. Ideally, this would be supplemented moreover with the numerical results (event counts) of the simulations.
  • validity: -
  • significance: ok
  • originality: ok
  • clarity: low
  • formatting: reasonable
  • grammar: good

Author:  Tanmoy Modak  on 2021-02-25  [id 1265]

(in reply to Report 2 on 2021-01-17)

  • Introduction and section 2:

  • As also critisized by the first referee, the model is not motivated well in the paper, nor is the considered parameter space. The reader is left to search the previous literature for details. The relevant references are mentioned throughout the text, but no consistent overview of the previous literature leading up to this work is given. The couplings lambda_ii and rho_ij are introduced completely ad hoc in the introduction; while some explanation comes in section 2, one has to make an effort to piece things together while reading.

We have now modified the introduction regarding our motivation. We have treated the extra Yukawa couplings as independent parameters and turned off all rho_ij couplings except for rho_tc for all practical purposes throughout the paper. We have also added discussions in the last paragraph of Sec 4 on the impact of turning on other rho_ij couplings.

  • Refs [3-5] are an extremely selective list of theorists' fits of the Higgs couplings. There exists a vast literature on this subject, most of which is ignored here. Moreover, and perhaps more importantly, dedicated measurements of the Higgs couplings have been published directly by ATLAS and CMS.

We have added a more comprehensive list of Run2 analyses.

  • Eq. (1) is duplicated in eq. (8).

We are aware of this, but in the interest of fast browsing we find this duplication useful.

  • It is not clear what are the actual constraints on rho_tt, and what are the consequences for this study if both rho_tt and rho_tc are relevant.

We added a discussion regarding this in the last paragraph of Sec.4.

  • Section 3

  • "For b-tagging as well as c-jet and light-jet rejection, we rely on Delphes" is too vague. The standard ATLAS card can change over time, so details on the assumed reconstruction efficiencies must be given, and not only for (b)jets but also for lepton ID. The same is true for extra jets: details on the matching and merging parameters are necessary.

We now mention the version of Delphes in its first instance i.e. second paragraph of Sec 3. We also mention now that we have used the default ATLAS Delphes card for b-tagging and light-jet rejection in the same paragraph.

  • Cross sections in Table 2: while it can be justified to compute sub-leading contributions at lower order, this needs to be justified and explained properly. "for simplicity" is not good enough.

We have accounted for the NLO (or NNLO) correction factors for all leading and subleading backgrounds. Only the subsubdominant backgrounds tWZ and ZZ+jets backgrounds are taken as LO, not because of simplicity, but because their experimental impact is unlikely to be detemined by QCD corrections. In addition, for all dominant backgrounds we have considered up to one additional jet in the final state. The number of additional jets are restricted to one for the different backgrounds due to coputational limitations.

  • How many events were actually generated?

For all signal and background events for both the charged and neutral Higgs processes we have generated "at least" 100k events. We have further made sure the number of surviving events after application of selection cuts are statistically significant.

  • Run 3 and HL-LHC reach: to what extent are the efficiencies of the standard ATLAS Delphes card appropriate for HL-LHC simulation?

Indeed such standard ATLAS Delphes card would likely be modified for HL-LHC. However as a first attempt we have utilized the current ATLAS based Delphes card of Delphes 3.4.2 for HL-LHC. A sizeable effects could be through the b-tagging rate combined with pile-up subtraction, but we admit that we really have no idea how to estimate this.

  • What about pile-up?

We have not considered pile-up in our analysis. Both the processes have leptonic final state and we expect the HL-LHC pile-up effect should not impact our results decisively (beyond the single b-tag).

  • Section 4

  • Same concerns as above for standard CMS Delphes card and details on the event generation.

Again we adopted similar strategy and used default CMS based detector card of Delphes 3.4.2.

  • The CMS 4 top analysis has been reinterpreted with MadAnalysis5 in Phys.Lett.B 784 (2018) 223 [1805.10835]. There, a specific Delphes card was developed to reproduce well the CMS results. This was for 35/fb of integrated luminosity, but an update of the recast code (and appropriately tuned Delphes card) are available also for the 137/fb version, see https://madanalysis.irmp.ucl.ac.be/wiki/PublicAnalysisDatabase While these recast codes consider only the signal regions of the CMS analysis but not the control regions, the fact that a specific tuning was necessary raises severe doubts about the validity of using the standard CMS Delphes card in the present work. How has this been validated? Details on this must be explained and the accuracy of the reinterpretation demonstrated.

We are aware of issues regarding the analysis of Ref.[100] (in current version). We would like to draw your attention to the acknowledgment. We indeed had to seek internal help and private communication for the details of the selection criteria mentioned in Eq.(19). Eq.(19) is not clearly mentioned not only in Ref.[100] but also its predecessor with up to 2016 data (Eur. Phys. J. C 78, 140 (2018)). As a first attempt we have not validated different control and signal regions of Ref.[100] and simply utilized default CMS based detector card of Delphes 3.4.2 in our analysis. This is primarily due to this reason we advocated a dedicated same sign-top search in the current manuscript and in Refs.[33,102].

  • The same-sign top search is a crucial for this work. Why is it relegated to the appendix? Again, relevant details are not explained.

While it indeed can probe the parameter space for |rho_tc|, however, the process has very mild dependence to c_gamma. However we have now elevated the appendix to the main text and with details of cut based analysis as suggested.

  • Given the very strong conclusions about ruling out the model under investigation, the impact of the uncertainties mentioned in the last paragraph before the conclusions should be quantified.

We had already briefly discussed on systematics however leave out a detailed analysis for future.

In conclusion, while the study is interesting in principle, much more detail (and care) is needed to make the paper suitable for publication. I actually have more comments on the paper, but given the shortcomings listed above I refrain from going into finer details in this iteration.

  • General remark: for the sake of reproducibility, the MadGraph model (UFO) file, all the Monte Carlo settings for the event generation (i.e. the MG5, Pythia, Delphes cards) and the analysis codes should be made public on an appropriate versioned and citable open-access repository, for example Zenodo. Ideally, this would be supplemented moreover with the numerical results (event counts) of the simulations.

Thank you. We indeed maintain a twiki page for the neutral Higgs processes such as cg > tA/tH > t t cbar for experimental usage. There indeed is a UFO file which is validated with an existing 2HDM model in FeynRules model database. The link is given below. We chose not to advertise it in the current paper since the charged Higgs processes are not incorporated into the UFO yet.

https://twiki.org/cgi-bin/view/Sandbox/FlavorChangingNeutralHiggs

---

## Round 1 · Referee Report · Anonymous (Referee 3) · 2021-2-3

Strengths

I believe that the paper "A Final Word on FCNC-Baryogenesis from Heavy Higgs Bosons" by Wei-Shu Hou, Tanmoy Modak and Tilman Plehn has enough novelty to be published in SciPost Physics.

  1. The model and parameter space are clearly formulated.
  2. It was shown the LHC can probe almost entire parameter space responsible for EWBG within the framework of two Higgs doublet model
  3. The the complicated 2HD model parameter space relevant to EWBG was elegantly expressed in terms of two variables -- |\rho| and |c_\gamma| and results a very clearly presented in this pane
  4. The phenomenological analysis are done at a good quality level using Madgraph-PYTHIA-Delphes chain including jet matching

To conclude -- the paper well written and contain new results of a good quality

Weaknesses

There are only few points which should be clarified before I could recommend the paper for the publication:

  1. Author have taken almost all relevant backgrounds for "b W^+W^+W^-" 3lepton signature, except "t t-bar" one: naively this background would lead only two di-lepton signature, however the third lepton could come from as "fake" one or from b-quark decay: the probability for this lepton is very low
  2. 0.1% - 0.01%, but as we know the tt-bar cross section is huge. So this background can be easily the leading one for the
    "b W^+W^+W^-" signature. It should bec checked or at least roughly estimated.

  3. I believe that title of the paper is too ambitious -- the paper is well addressing problem of probing of the parameter space only within the framework of 2HDM at the LHC, within the framework of more general/different model the limits in |\rho| and |c_\gamma| would be not necessarily correct or relevant. I think this should be clearly stressed in the conclusions and possibly in the title.

Report

I believe the paper deserve the publication in journal upon addressing problem mentioned above (and formulated in short below)

Requested changes

  1. Authors should estimate the contribution of ttbar background to 3lepton signature from "bH^+" (see details above)

  2. Authors should clearly states the validity of their results withih the prameter space of different models and possibly change the ttile which is more relevant to 2HDM

  • validity: good
  • significance: good
  • originality: good
  • clarity: good
  • formatting: good
  • grammar: good

Author:  Tanmoy Modak  on 2021-02-25  [id 1264]

(in reply to Report 3 on 2021-02-03)

  1. Author have taken almost all relevant backgrounds for "b W^+W^+W^-" 3lepton signature, except "t t-bar" one: naively this background would lead only two di-lepton signature, however the third lepton could come from as "fake" one or from b-quark decay: the probability for this lepton is very low - 0.1% - 0.01%, but as we know the tt-bar cross section is huge. So this background can be easily the leading one for the "b W^+W^+W^-" signature. It should bec checked or at least roughly estimated.

Thank you for pointing this out to us. We have now added the contribution from ttbar+jets background in Table 2. The discovery and exclusion limits have become a bit milder. The modified numbers are given in the second last and last paragraph of Sec.3. Correspondingly, Figures 4 and 5 are also modified.

  1. I believe that title of the paper is too ambitious -- the paper is well addressing problem of probing of the parameter space only within the framework of 2HDM at the LHC, within the framework of more general/different model the limits in |\rho| and |c_\gamma| would be not necessarily correct or relevant.I think this should be clearly stressed in the conclusions and possibly in the title.

We have modified the title and the last paragraph of Sec.5 to reflect this point. In addition, we have added discussions at other places and a new figure (Fig.1) as suggested by Referees 1 and 2.

---

## Round 2 · Referee Report · Anonymous (Referee 1) · 2021-4-13

Report

The authors have partially addressed my comments and criticism. I copy below my original report with their replies and my new comments, marked by `-->'.

model definition:

The potential is defined in eq. (4) and the couplings to fermions in eq. (6). It remains unclear what are the input parameters of the model. The various choices of the rho matrices have to be deduced from information scattered over the article. The choide of the elements of the rho matrices appears fully ad-hoc to satisfy experimental constraints. Is there any underlying model that would produce such a extremely weird pattern that the authors employ?

Our primary focus is to probe the parameter space required for rho_tc-EWBG. For practical purposes, we have turned off all rho_ij couplings except for rho_tc. We have added a discussion in the last paragraph of Sec.4 on the impact of turning on other rho_ij couplings. The dynamical parameters in the 2HDM potential would be subjected to constraints from perturbativity, positivity, unitarity, and oblique parameters. We have now added discussion regarding this in the first paragraph of page 4.

--> I understand that there is NO motivation for this weird choice of rho_ij, but just call it "rho_tc-EWBG". The added discussion at the end of Sec. 4 is interesting, but does not go very far.

checks on the model parameters:

There is hardly any information on the checks performed. The authors mention in passing perturbativity, positivity, unitarity, and electroweak precision data. It remains completely unclear what the effects of these are on the model as such and on the particular parameter choices the authors make. Completely absent seem to be checks for the properties of the SM-like Higgs boson at 125 GeV, as well as checks for the BSM Higgs searches for the additional Higgs bosons (except the particular channels the authors are analysing later).

We have added discussions and parameter scans (see Fig.1) for the allowed parameter space by perturbativity, positivity, unitarity, and electroweak precision data (checked via 2HDMC). As we turn off all rho_ij couplings, and given that rho_tc does not enter the Higgs signal strengths at tree level, the impact is minimal. However the mixing angle c_gamma would receive some constraints via hZZ, hWW etc. vertices. We had already checked that the chosen c_gamma range [0.1,0.3] is allowed by Higgs STXS measurements. We now added discussion in second paragraph of page 4.

--> It is positive that the authors have added (now) the checks for perturbativity, positivity, unitarity and electroweak precision data. The explanations concerning the Higgs signal strength measurements, however, remains vague. There are public tools such as Lilith or Higgssignals that facilitate such a test. This has not been done here, and how their bounds on c_gamma have been derived remains unclear. Furthermore, the authors still have not checked whether the BSM Higgs searches have an impact on their parameters (except the particular channels analysed).

Fig. 1 (now Fig. 2):

What is the point in showing it for MH+- = 300 GeV and 500 GeV? The only difference seems to come from Bs-Bsbar mixing. What are the other relevant parameters here? How do they influence the excluded regions? Furthermore, it appears that the searches for t -> ch can rule out the scenario under investigation in the (near?) future. The authors do not comment on this.

The plots for mH+ = 300 and 500 GeV are chosen for illustration. Along with the B_s mixing, the reach of cg > b H+ > b h W+ process is also different for the respective masses. In the previous version we inadvertently put the wrong 2sigma contour for the HL-LHC exclusion from cg > b H+ > b h W+ process in the left panel of Fig.4 (previous version). Please see the modified left panel Fig.5 in the latest version for comparision. We have added the projected HL-LHC reach for the t > ch decay, including comments in the first paragraph of page 5.

--> Fig. 2 still suggests that the "preferred range" is already largely ruled out, which is not discussed (apart from some general statements on the top of p. 5.) A minor point: why is mH+ = 300 GeV chosen for Fig. 2, but 350 GeV for the other figures?

"usual" BSM Higgs production and decay modes:

The authors are investigating solely the channels they are interested in. However, there should be some "usual" channels, such as gb -> tH+-, H+- -> tb, which may be suppressed (because of the weird and unmotivated choice of rho^U,D), but via CKM mixing they should still play a significant role. The authors do not comment on this.

Indeed, due to our choice of parameters such processes are not relevant and we have checked the constraints from searches such as gb -> tH+-, H+- -> tb are well within the experimental reach with our choice of parameters. We added discussions regarding this in the last paragraph of Sec 4.

--> As for previous items the authors make some "general statements", but not very substantial ones. It would be interesting to know what the more detailed results of the checks mentioned here have been. The role of the CKM matrix has also not been clarified.

Calculation of signal and background:

Some channels are calculated at NLO, others at LO, which in particular seems to include the signal channels. Why? I do not see any motivation for this from the physics point of view.

For the charged Higgs production cg > b H+ > b h W+ we have normalised our LO background cross sections upto NLO (at least) for all backgrounds, except for the SM tWZ background which is subsubdominant. Same is true for neutral Higgs production cg > tA/tH > t t cbar. In addition, we have considered at least one additional jets in the final state for all dominant backgrounds and signals. However, a full estimation of QCD corrections for the signal processes is beyond the scope of the current paper. Therefore we believe our discovery and exclusion contours are conservative.

--> I still fail to see the motivation for their choices. The authors also have not included their "explanations" into the text.

Fig. 4 (now Fig. 5):

It seems that what the authors label "CRW" alone is already excluding a lot of the allowed parameter space, so this should be looked at first, since it is an existing limit. Also this channel, which is not in the main focus of the authors, seems to be able to rule out their scenario with future LHC runs. Again this is not discussed.

The cg > tA/tH > t t cbar process can indeed probe a large part of the parameter space for |rho_tc|. However, we re-iteriate that the c_gamma dependence is very mild in cg > tA/tH > t t cbar, as can be seen from Fig.5. It is solely for this reason we put discussion on cg > b H+ > b h W+ process prior to neutral Higgs production cg > tA/tH > t t cbar. We have elevated the discussion in the appendix and merged it into the main text. As mentioned in the previous version (see also reply for Referee 2) the CRW region of Ref.[100] is not optimized for the cg > tA/tH > t t cbar process. Therefore we advocate a simple dedicated search for cg > tA/tH > t t cbar process as discussed in Sec.4.

--> The paper gains with the extended discussion of CRW. Still the fact that for mH+ = 350 GeV nearly the full parameter space is excluded, or who the relevance of this exclusion scales with the mass is not clearly discussed.

I do not include several smaller issues that I have, e.g. concerning

citations, since the above issues are far more relevant. But looking at Ref. [84] I wonder whether the authors really read their article before submitting it.

We use its estimate of the NLO SM ttbarh cross section to determine the QCD correction factor for SM ttbarh background. We are not entirely sure what the problem is and why this has to be phrased in a rather unprofessional and not even funny manner.

--> The authors now at least included "twiki:" into the reference (now [96]), which besides that solely consists of the title of that twiki page, but without any URL. The accessibility of the URL depends on the details/configuration of the pdf reader.

In summary, the analysis lacks motivation (the authors do not provide any for only considering rho_tc, which looks completely artificial) and partially also the rigorousness I would expect for a published article.

  • validity: low
  • significance: low
  • originality: ok
  • clarity: ok
  • formatting: -
  • grammar: -

Author:  Tanmoy Modak  on 2021-05-12  [id 1419]

(in reply to Report 1 on 2021-04-13)

--> I understand that there is NO motivation for this weird choice of
rho_ij, but just call it "rho_tc-EWBG". The added discussion at
the end of Sec. 4 is interesting, but does not go very far.

The 2HDM is one of the most attractive models for electroweak
baryogenesis, which in turn is a key motivation for BSM physics. We
include CP-violation through the Yukawa sector, leading us to this
specific choice of rho_tc. We changed the discussion to make this more
clear - our model is directly motivated through electroweak
baryogenesis and minimal particle content.

--> It is positive that the authors have added (now) the checks for
perturbativity, positivity, unitarity and electroweak precision
data. The explanations concerning the Higgs signal strength
measurements, however, remains vague. There are public tools such
as Lilith or Higgssignals that facilitate such a test. This has
not been done here, and how their bounds on c_gamma have been
derived remains unclear. Furthermore, the authors still have not
checked whether the BSM Higgs searches have an impact on their
parameters (except the particular channels analysed).

We feel quite confident with our BSM-Higgs and global Higgs expertize
in Heidelberg. At tree level, the combination of rho_tc and c_gamma
only affects t > ch, as discussed. In addition, c_gamma affects the
hZZ and hWW vertices, and we have checked the current limits. We are
of course happy to check additional searches, if there are any missing.

--> Fig. 2 still suggests that the "preferred range" is already
largely ruled out, which is not discussed (apart from some general
statements on the top of p. 5.) A minor point: why is mH+ = 300
GeV chosen for Fig. 2, but 350 GeV for the other figures?

We added a discussion on this after (now) Eq.(12). In short, even an
observation of the anomalous top decay would not provide the necessary
insight into baryogenesis. The choice of mass in Fig.2 is driven by a
faithful representation of Ref.[75]. In the rest of the paper we
re-estimate the B_s constraints for mH+ = 350 GeV and 500 GeV and now
say that in the text.

--> As for previous items the authors make some "general statements",
but not very substantial ones. It would be interesting to know
what the more detailed results of the checks mentioned here have
been. The role of the CKM matrix has also not been clarified.

We are a little baffled by this "general" comment. We have slightly
modified the introduction to clarify the role of the CKM matrix, it is
also properly defined in Eqs.(7,8).

--> I still fail to see the motivation for their choices. The authors
also have not included their "explanations" into the text.

The choice is driven by the goal to include higher-order corrections
to the total rate whenever they are available in literature and might
be relevant for our analysis. The number of extra hard jets ensures
that we are not affected by parton shower approximations. We are, of
course, happy to check if an available higher-order prediction affect
our results. We find this procedure quite standard.

--> The paper gains with the extended discussion of CRW. Still the
fact that for mH+ = 350 GeV nearly the full parameter space is
excluded, or who the relevance of this exclusion scales with the
mass is not clearly discussed.

As mentioned above, we added an improved motivation of our proposed
search for the new Higgs states (related to baryogenesis) to the
discussion of Fig.2.

--> The authors now at least included "twiki:" into the reference (now
[96]), which besides that solely consists of the title of that twiki
page, but without any URL. The accessibility of the URL depends on the
details/configuration of the pdf reader.

We are happy to improve our LaTeX, if the referee tells us which
viewer has issues with the hyperlink. We are using standard bibtex.

--> In summary, the analysis lacks motivation (the authors do not
provide any for only considering rho_tc, which looks completely
artificial) and partially also the rigorousness I would expect for a
published article.

We are a little surprised by the referee insisting in this point of
view, because we would have considered electroweak baryogenesis a fine
motivation for LHC analyses. In general, we would like to emphasize
that we made a serious effort to accommodate *all* comments by the
referee and changed parts of the paper whenever they appeared
unclear. Finally, we went through the text once more and streamlined
some of the discussions after considering all referee comments.

Attachment:

manuscript_v3_MPETkFw.pdf

---

## Round 2 · Referee Report · Anonymous (Referee 2) · 2021-4-27

Report

The paper has been somewhat improved with respect to the original submission, but still does not give all the necessary details and overall seems to lack scrutiny. Several of the points raised in my first report are addressed in an unsatisfactory manner. Concretely:

  • Refs [3-11] (previously Refs [3-5]) : the list is somewhat more comprehensive now, but really does not justice to the experimental efforts. The text says "measurements of the Higgs Lagrangian" so one should expect first of all the citation of the relevant experimental papers and only second the theorists' fits. This may seem like a minor point, but I see no justification for ignoring the ATLAS+CMS Run 1 combination paper at this point.

  • Eq. (1) is duplicated in eq. (9) (previously eq. (8)): I would buy the argument about "fast browsing" in a long review, but not in a ~10 pages article. This is only a stylistic point, but eq. (9) would better be integrated in the text, with reference to eq. (1).

  • Constraints on the parameter space mentioned in the two paragraphs after eq. (10): these explanations are completely insufficient. It has to be explained clearly and unambiguously how these constraints are applied (or derived, in case of c_gamma) . Since this was also criticised by Referee 1, I do not go into further detail here.

  • Cross sections in Table 2: the visible cross sections of tth, tZ+jets and tWZ are of comparable size, so it is not true that tWZ is subsubdominant. Moreover, the largest single contribution is taken at NLO while smaller contributions are taken at NNLO. This has to be justified (or improved).

  • My biggest concern is still the usage of the standard CMS Delphes card for the reinterpretation of the CMS 4 top analysis without any proof that this actually allows to reproduce the results of the CMS analysis with good enough precision. Such a reinterpretation without any validation is simply not acceptable.

  • A constraint on rho_tt is now given in the last paragraph of Sec. 4, but only hand-waving arguments are given regarding the consequences of non-zero rho_tt. It is not clear at all how the bounds derived in this study would change both rho_tt and rho_tc were relevant.

All in all, the conclusion from my first report that much more detail and care would be needed to make the paper suitable for publication still holds. The journal's acceptance criteria are not met by this study and I therefore cannot recommend it for publication in SciPost Physics.

  • validity: -
  • significance: -
  • originality: -
  • clarity: -
  • formatting: -
  • grammar: -

Author:  Tanmoy Modak  on 2021-05-12  [id 1418]

(in reply to Report 2 on 2021-04-27)

  • Refs [3-11] (previously Refs [3-5]) : the list is somewhat more comprehensive now, but really does not justice to the experimental efforts. The text says "measurements of the Higgs Lagrangian" so one should expect first of all the citation of the relevant experimental papers and only second the theorists' fits. This may seem like a minor point, but I see no justification for ignoring the ATLAS+CMS Run 1 combination paper at this point.

There exist a vast number of combined and global analyses for the LHC Run 1, so in the interest of a finite list we compiled a comprehensive list of global Run 2 analyses. We changed the order of the references to first cite ATLAS and CMS.

  • Eq. (1) is duplicated in eq. (9) (previously eq. (8)): I would buy the argument about "fast browsing" in a long review, but not in a ~10 pages article. This is only a stylistic point, but eq. (9) would better be integrated in the text, with reference to eq. (1).

We appreciate the referee's point, but given that SciPost does not have page limits we decided to rather repeat this central piece of information for advanced readers who (rightfull) skip the introduction.

  • Constraints on the parameter space mentioned in the two paragraphs after eq. (10): these explanations are completely insufficient. It has to be explained clearly and unambiguously how these constraints are applied (or derived, in case of c_gamma) . Since this was also criticised by Referee 1, I do not go into further detail here.

We expanded and slightly re-ordered the discussion and hope that it is now clear.

  • Cross sections in Table 2: the visible cross sections of tth, tZ+jets and tWZ are of comparable size, so it is not true that tWZ is subsubdominant. Moreover, the largest single contribution is taken at NLO while smaller contributions are taken at NNLO. This has to be justified (or improved).

We addressed this issue in the response to Referee 1. We also include available NNLO predictions when available, to make sure that our predictions are as precise as possible. The same holds for the extra number of hard jets. If we are leaving out a NNLO correction for a relevant but complex background, it is because it is not available. We are, of course, happy to include further K-factors in case we miss them.

  • My biggest concern is still the usage of the standard CMS Delphes card for the reinterpretation of the CMS 4 top analysis without any proof that this actually allows to reproduce the results of the CMS analysis with good enough precision. Such a reinterpretation without any validation is simply not acceptable.

We agree that this approach is not acceptable, in the sense that the standard Delphes card should not be the best available detector simulation. Agreeing with the referee on this aspect, we sought expert advice to confirm the validity of our simulations. Because this constraint is only a side aspect of our paper, we refer to Ref.[100] with a large overlap in authorship for details. In any case, we argue that our neutral channel in Eq.(18) is most likely a more robust probe of the parameter space.

  • A constraint on rho_tt is now given in the last paragraph of Sec. 4, but only hand-waving arguments are given regarding the consequences of non-zero rho_tt. It is not clear at all how the bounds derived in this study would change both rho_tt and rho_tc were relevant.

As we are (now) discussing in some detail on p.10, rho_tt would dilute our signature for mA,mH > 2 mt ( mH+- > mt + mb) by supressing the decays A/H > t tbar and H+- > t bbar/tbar b. However, combining rho_tt and rho_tc would induce novel cg > tA/tH > t t tbar and cg > bH+ > b t bbar signatures. These searches are already performed and discussed in detail in Refs.[32,39].

Attachment:

manuscript_v3.pdf

---

## Editorial Decision

published